# Current Cognition Tests, Potential Virtual Reality Applications, and Serious Games in Cognitive Assessment and Non-Pharmacological Therapy for Neurocognitive Disorders

**DOI:** 10.3390/jcm9103287

**Published:** 2020-10-13

**Authors:** Roger Jin, Alexander Pilozzi, Xudong Huang

**Affiliations:** Department of Psychiatry, Massachusetts General Hospital and Harvard Medical School, Charlestown, MA 02129, USA; rogerjin@mit.edu (R.J.); apilozzi@mgh.harvard.edu (A.P.)

**Keywords:** aging, major neurocognitive disorder, Alzheimer’s disease, cognitive assessment, mild neurocognitive disorder, mini-mental state examination, virtual reality, augmented reality, mixed reality

## Abstract

As the global population ages, the incidence of major neurocognitive disorders (major NCDs), such as the most common geriatric major NCD, Alzheimer’s disease (AD), has grown. Thus, the need for more definitive cognitive assessment or even effective non-pharmacological intervention for age-related NCDs is becoming more and more pressing given that no definitive diagnostics or efficacious therapeutics are currently unavailable for them. We evaluate the current state of the art of cognitive assessment for major NCDs, and then briefly glance ahead at potential application of virtual reality (VR) technologies in major NCD assessment and in cognition training of visuospatial reasoning in a 3D environment, as well as in the alleviation of depression and other symptoms of cognitive disorders. We believe that VR-based technologies have tremendous potentials in cognitive assessment and non-pharmacological therapy for major NCDs.

## 1. Introduction

The present review seeks to define and describe some common tests for screening mild neurocognitive disorders (mild NCDs) and major NCDs. Tests were selected based on their use and the areas of cognition they cover. Furthermore, the review surveys the potential for virtual reality and related technologies in major NCD screening, as well as for the amelioration of cognitive deficiencies in cases of cognitive impairment and in the general older population.

### 1.1. Normal Aging

In a review on diagnosis of major NCDs, it is natural to first consider the state of normalcy for comparison. Cognitive decline that occurs with normal aging is well characterized. Age brings defects in many areas of cognition. It appears that while “crystallized intelligence”—cognition involving learned or practiced information or procedures—is resistant to decline with age, “fluid intelligence”—cognition involving reasoning with unfamiliar ideas or information—tends to become impaired with age. Briefly, we consider some facets of fluid intelligence that are affected. One afflicted area of cognition is the encoding of knowledge into memory. A 2003 cross-sectional study by Haaland et al. examining recall and recognition ability in the context of logical memory and visual reproduction suggested that the largest factor contributing to learning deficits in aging was the encoding of information into memory, rather than retrieval [1].

Another significant facet of cognition is selective attention—the ability to both focus on a point of interest as well as to filter out irrelevant information. Using the reading-with-distraction task as a metric, in which subjects are instructed to read several passages while in the presence of external disturbances, Darowski et al. found in a cross-sectional study on patients aged 18–87 years that it appears that selective attention declines with age [2].

A third area of cognition is logical and spatial reasoning. A cross-sectional study on 120 males of ages 20–79 conducted by Salthouse et al. found that both verbal and spatial reasoning performance declines with age [3].

It is worth noting, however, that while cross-sectional studies of cognition across many ages can obtain results quickly, the groups were raised and educated in different eras, an unfortunate reality that may greatly influence test performance. Indeed, a Brazilian study published in 2015 found that, among 14,594 persons aged 35–74 years, participants’ education level had a higher influence over cognitive test scores than did their age [4]. Furthermore, when age-related changes are measured through longitudinal and cross-sectional studies, results are often different; cross-sectional analyses are impacted by differences between study cohorts, while longitudinal analyses introduce factors such as experience with the testing protocol, thus confounding results [5].

### 1.2. Mild Neurocognitive Disorder

Although cognitive abilities decline with normal aging, slightly accelerated states of cognitive decline exist and are given the classification of mild cognitive disorders (mild NCDs) as an intermediate state between normal aging and major NCDs. A primary concern of the prognosis of a mild NCD patient is progression into a major NCD, though the etiology of mild NCDs is highly heterogenous, and not all individuals exhibiting mild NCDs will develop major NCDs; many of those with mild NCDs will even revert to cognitive normalcy [6,7]. However, individuals with mild NCDs have a significantly higher risk of developing major NCDs [8,9], particularly in cases of the amnestic mild NCD subtypes [10]. In practice, the feature that distinguishes mild NCDs from major NCDs is the interference of cognitive decline with the performance of daily activities, such as cooking or self-hygiene, or social functions [11].

### 1.3. Alzheimer’s Disease

Alzheimer’s disease (AD), a degenerative neurological disorder characterized by deficits in memory and spatial reasoning, is the most common form of major NCD and, in most cases, does not have a well-established underlying cause or treatment. In addition to cognitive deficits, AD is characterized by intrasynaptic β-amyloid plaques and intraneuronal τ-protein neurofibrillary tangles, although it is unknown whether either of these patent markers are causal in the disease etiology [12]. AD is also characterized by degeneration of cholinergic neurons on the basal forebrain. AD most commonly affects patients over 65 years of age, although familial AD can begin affecting patients well below the age of 65. Around 10% of those age 65 and older in the US have a major NCD in the form of AD, and the frequency increases with age; 43% of those over the age of 85 have the disorder [13]. A single definite cause has not been identified, though there is association with certain genetic mutations and other abnormalities, such as trisomy 21 [14,15,16].

Because AD has well-studied physical manifestations in the brain, risk for AD can be assessed decades before the potential onset of the disease through advances in neuroimaging. This implies that unlike normal aging or mild NCDs, which currently must be diagnosed through cognitive assessments, the AD state has the potential of a reference-standard disease state at autopsy, when the brain can be sectioned and examined for amyloid deposits and neurofibrillary tangles. However, because longitudinal studies are expensive and slow, magnetic resonance imaging (MRI) and other imaging data must usually suffice as a reference standard. This also implies that for patients who have already been diagnosed with preclinical signs of AD, the conversion from mild NCD to AD is a bit contrived because diagnostic criteria for AD can be discerned decades before cognitive symptoms appear [17]. Thus, in some cases, mild NCDs are an intermediate diagnosis as the patient progresses into AD or major NCDs more generally.

## 2. Motivation for Cognitive Assessment

Uncertain boundaries between what is currently considered normal age-related decline and noteworthy cognitive impairment, alongside a lack of reliable biomarkers for major-NCD-causing diseases, make the process of diagnosis somewhat difficult. Though MRIs and other methods can identify notable aberrations in brain health, they are neither easy nor foolproof to administer. Cognitive assessments fill the gap, providing a relatively easy, non-invasive means of assessing one’s cognitive status. In practice, we find that the current amount of diagnostic information garnered from MRI data is not a substitute for that which can be learned from cognitive assessment. Lebedeva et al. found that adding Mini-Mental State Examination (MMSE) input to a random-forest classifier improved test accuracy, sensitivity, and specificity compared to using MRI data alone [18]. Major NCDs are also not homogenous conditions. Though AD is the most common cause of major NCDs, it is not the only cause [13,19]. Though AD accounts for approximately 60–80% of major NCD cases, other forms of major NCDs, such as Vascular, Parkinson-related, and frontotemporal major NCDs are prevalent [19]. Assessments that target the cognitive symptoms of major NCDs are useful for screening patients for further, more in-depth analysis due to the lack of a catch-all physiological marker for all major and minor NCD variants. Indeed, the current Diagnostic and Statistical Manual of Mental Disorders (DSM)-5 criteria define minor NCDs as the presence of a marked decrease in cognitive ability, as noted by an informant or a cognitive test that does not yet impede the subject’s day-to-day life and that is not explained by delirium or a mental disorder [20].

### 2.1. Definition of an Effective Cognitive Assessment

Before considering the particulars of any one cognitive assessment, it is useful to list characteristics that are important to the development of cognitive assessments. We segregate these properties into practical needs, which deal with the details of administration, and diagnostic needs, which deal with the ability to discern the subject’s cognitive status.

#### 2.1.1. Practical Needs

The test should be brief. Patients with major NCDs often suffer from deficiencies in attention; thus, a long test may not be suitable for such patients. A brief test is also more convenient for the clinician to administer, and more manageable in terms of health-care capacity.The questions of the test should not lead the patient to experience emotional distress.The test should ideally avoid involvement of the patient’s family and collaterals.The test should be easy to administer, requiring minimal training to give and score.The test should be cheap and minimalistic in terms of materials required.

#### 2.1.2. Diagnostic Needs

The test should have high sensitivity and specificity with respect to distinguishing between the normal and diseased states.High sensitivity/specificity with respect to change or progression of major NCDs: This implies that the test should be repeatable. Patients should not be able to easily memorize the answers to the test questions. This also implies that the maximum score on the test should be high, and the mean score of those who test positive should be around 50% of the maximum possible to facilitate separation of results.The test should cover various areas of cognitive function.The test results should be consistent and independent of the administrator and rater.The test should have minimal ceiling and floor effects. That is, the scores should not be clustered around the minimum or maximum possible scores. In terms of test design, the questions on the test should span a large range of difficulty.

## 3. Common Cognitive Assessments

### 3.1. Mini-Mental State Examination

#### 3.1.1. Background

The Mini-Mental State Examination (MMSE) is currently the most popular cognitive assessment. It consists of 11 questions totaling 30 points used to categorize the degree of a subject’s cognitive impairment from cognitive normalcy to severe impairment. The questions assess the cognitive areas of orientation, registration, attention, calculation, recall, and language [21]. The cutoffs for cognitive normalcy are generally varied based on education level, as the standard cutoff score of around 23 must be raised to achieve similar sensitivity/specificity when testing highly educated subjects [22]. Many variants of this test have arisen, some adapted to patients with special needs. Some variants include standardized MMSE, a more scripted version of the MMSE, the modified MMSE (3MS), which broadens scoring to 100 points and adds a few questions to test more areas of cognition, and hearing-/vision-impaired MMSE [23].

#### 3.1.2. Advantages

Perhaps the largest advantage of the MMSE is its popularity. Because it is the most popular cognitive assessment, data from a large patient pool are available for analysis for calibration of test scoring and comparison between patients. Another advantage of the MMSE is its simplicity. The MMSE requires minimal training to administer and score, and can be completed in under 10 min. Despite its simplicity, the MMSE has relatively high accuracy and specificity with respect to identifying patients with major NCDs. A review of data including MMSE scores from 1141 persons who had 16+ years of education found that the MMSE had an accuracy of 89%, sensitivity of 66%, and specificity of 99% when used to distinguish between educated patients who were either normal or had relatively severe major NCDs [22]. In the same vein, despite its simplicity, the MMSE targets a broad range of cognitive functions, with each area encompassing at least three points of the test. This allows for both a fairly broad assessment of a patient’s cognition as well as an initial screen for the areas of cognition in which the patient suffers the greatest deficit. Most impressively, perhaps, scores on the MMSE were found to be correlated with morphological attributes of the brain, especially of the hippocampus [24], notably involved in episodic and spatial memory [25], as well as the amygdala, cingulate gyrus, and parahippocampal gyrus [24], meaning that the MMSE yields similar information to much more expensive tests.

#### 3.1.3. Disadvantages

Currently, one disadvantage of the MMSE is its associated intellectual property issues. Although the material cost of administration is, in theory, negligible because the MMSE is patented, clinicians must pay royalties to MiniMental, the current patent holders, for every instance of MMSE administration, at least for the official, up-to-date version. This inconvenience has served as an impetus for the development of royalty-free cognitive assessments [21]. Another disadvantage of the MMSE is that while it focuses on several different areas of cognition, it is limited in its assessment of other important aspects, such as visuospatial reasoning, for which there is only a single question. This is a critical shortcoming of the test because a deficit in visuospatial navigation is a classic symptom of AD, as well as amnestic mild NCDs [26]. The test is also heavily language-based, which can confound results if the subject’s language ability and level of literacy are low [27,28,29,30]. Furthermore, the assessment has a relatively low sensitivity, necessitating secondary assessment for patients who are able to attain relatively high scores on the MMSE and yet exhibit symptoms of major NCDs [22]. In addition to the low sensitivity, the test is limited in its ability to predict future decline; however, analytical tools, such as COmputational Model to Predict the development of Alzheimer’s diSease Spectrum (COMPASS), are being developed to remedy this shortcoming [31].

### 3.2. Clock-Drawing Test

#### 3.2.1. Background

The clock-drawing test is one of the more conceptually simple of the cognitive tests in common use. Though there are a variety of scoring systems that have been applied to the test, they generally assign points to different elements of the clock, such as its shape, and how the numbers and clock hands are positioned [32]. The patient is assigned the task of drawing an analog clockface at a certain time, usually 11:10. This particular time is chosen because it forces the use of both visual fields as well as the inhibition of “frontal pull” of the number 10. A common mistake in major NCD patients is to draw the minute-hand pointing to the number 10 on the clockface rather than the number 2 [33]. The resistance of this allure is a sign of mental flexibility. There are various rubrics for scoring the clock, but the classic one scores on a 10-point scale and focuses on visuospatial ability [34].

#### 3.2.2. Advantages

Perhaps the main advantage of the clock-drawing test is its simplicity. The test can be administered with minimal training in under two minutes. Another significant advantage of the test is its ability to test multiple areas of cognition, especially visuospatial reasoning. The test assesses praxis through the act of drawing fine details. The test assesses calculation through the need to position the minute hand at the correct time. The test assesses visuospatial reasoning in the relative positioning and sizing of the numbers of the clock face.

#### 3.2.3. Disadvantages

The main disadvantage of the clock-drawing test is inconsistency of scoring. Because the prompt is so open-ended, the scoring may vary from rater to rater. In addition, because the test is so simple, the test suffers from floor and ceiling effects; patients with major NCDs may be unable to draw even the circle of the clock face, while some patients with only mild NCDs may be able to draw the clock perfectly. These two groups of patients will require secondary assessment to separate. Next, the simplicity of the clock-drawing test means it excludes some significant areas of cognition, particularly those related to language. These three disadvantages suggest that the clock-drawing test may work best as a component of a larger test, so as to minimize the effect of variation in its scoring on the total score as well as to mitigate floor and ceiling effects and broaden the areas of cognition tested. Lastly, this test may be biased against those who have not received formal education. For example, a Brazilian study of confounding factors that may influence a subject’s performance on the clock-drawing test found a correlation between education level and performance on the clock-drawing test [35].

### 3.3. Addenbrooke’s Cognitive Examinations

#### 3.3.1. Background

Addenbrooke’s Cognitive Examination (ACE) was developed as an extension of the MMSE to incorporate questions that tested areas of cognition not covered by the MMSE, such as visuospatial reasoning. The MMSE is incorporated verbatim into the assessment so that a 30-point MMSE sub-score might be generated. The ACE has been the foundational work for many different variations—including the ACE-Revised (ACE-R), developed to have clearly defined subdomain scores, and the ACE-III, developed to reinforce certain areas of the ACE-R, such as repetition and comprehension, and replaces the MMSE portion of the ACE and ACE-R [36]. The typical cut-off for major NCDs is 82–88 out of 100 points of the composite score for the ACE-III [36].

#### 3.3.2. Advantages

One of the advantages of the ACE is its ability to test visuospatial capability in both 2D and 3D, incorporating such challenges as reassembling fragmented letters and spatial reasoning with a wire Necker cube, while preserving the plethora of cognitive areas tested by the MMSE. Because it fills several holes left by the MMSE, the ACE outperforms the MMSE in diagnostic ability. The creators of a 30-point simplified version of ACE, known as the mini-ACE (M-ACE), compared their test to the MMSE, and found that the M-ACE was both more sensitive and less prone to ceiling effects than the MMSE. It was found that patients scoring below 21/30 points on the M-ACE almost certainly had a major NCD [37]. The ACE is also less prone to floor effects. A 2000 study validating ACE’s ability to differentiate mild or major NCDs, AD, and frontotemporal major NCDs found that the test was able to distinguish between different stages of Alzheimer’s disease [38]. Another advantage of the ACE is that administration is easy and requires minimal training. Lastly, it appears that the test remains effective when translated into other languages. For example, a Japanese version of the ACE achieved a sensitivity of 0.889 and a specificity of 0.987 using a threshold of 74/100 [39].

#### 3.3.3. Disadvantages

Aside from the ACE-III, which is independent of the MMSE, the versions of the ACE that contain questions from the MMSE are subject to the same patent issues. In addition, since the ACE was developed in the United Kingdom (UK), some of the questions are based on knowledge of the culture and politics of the UK, making it biased towards those with cultural knowledge of the UK or other European countries with similar political systems. For example, some of the questions ask for the name of the prime minister or leader of the opposing political party [40].

### 3.4. General Practitioner Assessment of Cognition

#### 3.4.1. Background

The General Practitioner Assessment of Cognition (GPCOG) was developed for use by doctors in a primary care setting. The test format consists of two interviews—one with the patient and one with an informant, a close friend or relative of the patient. The patient interview, which consists of nine questions, can alternatively be completed as a written or online form [41]. The patient interview incorporates the clock-drawing test as well as recollection of an address and a recent event in the news. The informant interview consists of six questions surrounding the patient’s ability to perform day-to-day tasks, such as finding objects, speaking, and managing finances [41].

#### 3.4.2. Advantages

In just nine questions directed to the patient, the GPCOG tests the areas of orientation, visual spatial abilities, executive function, retrieval of recent information, and delayed recall [41]. Despite being shorter than the MMSE, taking about four minutes for the patient interview and two minutes for the informant interview, the test has comparable sensitivity and specificity [41]. The written and online options for the patient assessment also free the physician to perform other duties during this time. In addition, while the MMSE is patented, the GPCOG is free to use. Lastly, it appears that the GPCOG is agnostic to cultural background and education level [42].

#### 3.4.3. Disadvantages

One disadvantage of the test is that it is reliant on the presence of an informant. Additionally, while the patient’s age was not a factor in the informant-based section of the GPCOG, the age was found to be a significant (corr = −0.187, *p* < 0.01) predictor of their score on the cognitive assessment portion [43].

### 3.5. Montreal Cognitive Assessment

#### 3.5.1. Background

The Montreal cognitive assessment (MoCA) is a screening tool developed to be an alternative to the MMSE that is better suited to the detection of mild NCDs, with the initial test achieving better sensitivity for both mild NCDs and AD, albeit with lesser specificity [44]; with tuning of the cutoff scores, however, the MOCA can achieve better sensitivity and specificity in detecting mild NCDs than the MMSE [45]. It can be used in conjunction with the MMSE to assess persons who fall within the normal cognitive range of the MMSE score, but display other signs of cognitive impairment, such as difficulty completing activities of daily living, or used to test those whose disorder is mild enough that their daily lives are unaffected. The test is paper-based and free to use. It incorporates questions and other cognitive tests that address a wide variety of common domains. A modified trail-making test assesses visuomotor, visuoperceptual, and task-switching ability, and the clock-drawing test is included as well. Other assessments in the MoCA examine language ability, conceptual thinking, recall/memory, and orientation [46].

#### 3.5.2. Advantages

The test was designed to take around 10 min to administer, and it is short and easily administered in a standard clinical setting, though it also assesses many important areas of cognition. The test is freely distributed as well [46]. The MoCA has also been found to indicate one’s cognitive reserve with greater sensitivity than the MMSE, which may explain, in part, the MoCA’s heightened sensitivity to early-stage AD [47]; individuals with greater cognitive reserves have been found to be at lower risk of developing major NCDs [48].

#### 3.5.3. Disadvantages

A notable disadvantage of the MoCA relates to sensory impairment. In a study comparing the MoCA scores of 301 older persons, half of whom had no sensory impairments, while 38%, 5%, and 7% were hearing, vision, or dual-sense impaired, respectively, it was found that, despite modifying the scores of the sensory-impaired persons, those who had no impairments were more likely to pass [49]. The test also needs to be corrected for age and education level; although, as with other tests, the total score is taken and compared to a cutoff, there were certain questions with substantially higher failure rates among 2653 older persons with cardiovascular disease [50].

### 3.6. Summary of Advantages/Disadvantages

Some of the key advantages and disadvantages of the tests we have discussed are summarized in the Table 1 below.

## 4. Applications of Virtual Reality in Cognitive Disorder Testing and Therapy

### 4.1. Virtual Reality Diagnostics

#### 4.1.1. Background

As a concept, virtual reality (VR) involves immersing an individual in a simulated environment. Users are able to interact with this environment through some set of physical sensors, and typically perceive that environment through a headset that provides visuals and audio for the environment; in some cases, tactile feedback can be transmitted to the user through other systems, such as controllers [51]. The unique features of VR systems have generated considerable interest in the medical/psychiatric fields; within the scope of major NCDs, the allure from VR comes from its ability to assess and improve spatial reasoning, along with other cognitive abilities, depending on the focus and design of the program [51,52].

The deterioration of visuospatial reasoning is a marker of Alzheimer’s and even amnestic mild NCDs [26]. One flaw in the current state-of-the-art diagnostic examinations for major NCDs is that, while it is possible to test spatial cognition with simple visuospatial tests, including some paper-and-pencil methods, these tests tend to correlate only partially, if at all, with spatial navigation in large-scale environments [53]. One potential method for incorporating large-scale spatial reasoning tasks in 3D is setting the test environment in virtual reality. The navigation of virtual reality environments has been found to correlate with their real-world counterparts, with the ability to safely navigate them in correlation with MMSE score [54].

#### 4.1.2. Egocentric and Allocentric Navigation

There are two broad categories of navigation strategies—egocentric, in which the individual navigates by considering objects relative to his or her location, and allocentric, in which the individual navigates by considering the positioning of objects relative to each other. Weniger et al. characterized deficits in egocentric navigation in Alzheimer’s and mild NCDs by asking patients to memorize a VR park environment. The authors furthermore characterized deficits in allocentric navigation by asking patients to learn a VR maze. The study failed to differentiate those with mild NCDs that would progress into major NCDs from those whose mild NCDs did not convert, but mild NCD individuals performed worse than controls on both egocentric and allocentric navigation tasks [55]. Serino et al. similarly investigated the allocentric and egocentric spatial abilities by having amnestic mild NCD, AD, and control participants enter a virtual room containing certain objects, including a target object. In the second phase, one of two tests was conducted. In the first, patients were presented with a top-down representation of the room without the target object, but with other objects present. In the second, patients entered a second VR room that was empty, with only an arrow for directional orientation; patients entered this room from a different side from the original. Significant differences were found between AD patients and controls, though these differences were dependent on both the task and the direction from which participants entered the virtual room [56]. Cushman et al. tested the ability of Alzheimer’s, mild NCD, and normal older individuals to navigate virtual and real-world environments, testing navigation and recall abilities. Both the real-world and virtual tests were effective in differentiating between groups, and, notably, there was no significant difference between real-world and VR performances [57].

#### 4.1.3. Navigational Memory

Lee et al. created a non-immersive virtual version of the classic radial arm maze (RAM) to test spatial working memory. The maze consists of six arms, one of which contains a goal, attached to a central platform. In order to visit an arm after exiting another arm, one must first visit the central platform. The primary challenge of the maze is to remember which arms have been visited and which have not. The maze can be revisited, and the mazes are designed so that some of the arms will never have treasure. The act of learning and recalling which arms will never hold treasure is termed “spatial reference memory”. Lee et al. found that in AD patients, both spatial reference and working memory were impaired, while in mild NCD patients, only spatial reference memory was impaired [58]. Plancher et al. investigated the episodic memories of amnestic mild NCD patients, AD patients, and controls by challenging them to both active and passive vehicular exploration of a virtual reality city environment. Participants were tasked with recalling details about the route, including allocentric and egocentric navigational recollection (e.g., remembering the position/sequence of elements). The cognitive status of the participant was a significant indicator of impaired performance, though this effect was much greater for AD patients than those with amnestic mild NCDs [59].

#### 4.1.4. Activities of Daily Living (ADL)

One’s ability to complete tasks that are part of daily life is integral to the definition of major NCDs, which are classified as cognitive impairments that make completing activities of daily living difficult [60]. However, using this criterion in diagnosis is difficult; though a wide variety of surveys and questionnaires have been developed, they rely on interviews with a patient and/or caregiver, which can be subjective and influenced by the subject’s interpretation of the specific wording of a question; additionally, reliable administration is largely limited to outpatient settings [61,62,63]. VR notably allows for the recreation of tasks a participant might reasonably encounter, so that participants can be scored on their ability to complete them. Tarnanas et al. used a non-traditional virtual reality setup featuring a treadmill and curved screen to simulate a “day out” task to screen for early major NCDs; participants were tasked with navigating fire-building scenarios of varying degrees of difficulty. Individuals’ performance scores were found to be a better predictor of mild NCD to AD conversion than the Bristol ADL and Blessed ADL tests [64]. Dulua et al. created a VR experience called “A Day to Remember”, which featured daily-life activities as well as gamified cognitive tests. Most participants found the VR game to be easy to use, and noted that the experience was less stressful compared to the direct screening tests. However, while the test was functionally correlated to the paper MMSE, its precision was somewhat lacking, and a participant’s difficulty controlling the new technology was suggested to be a confounding factor [65]. Zygouris et al. created a low-immersion virtual supermarket, which measured aspects of task completion, such as the time to complete a shopping list and the accuracy of the items bought, money paid, and types of items bought. All participants were able to complete the exercise and were able to self-administer the test at home via a provided tablet [66,67]. An initial study used the aforementioned performance metrics in order to differentiate 34 mild NCD patients from a sample of 55. A decision tree was constructed based on the mistakes made and certain age/performance thresholds. A sensitivity of 82% and a specificity of 95% was achieved. In a secondary study using the same method but applying data on six healthy and six mild NCD individuals to a Naïve-Bayes classifier using the data from 20 of the participants’ trials, the classifier achieved a sensitivity of 94% and a specificity of 89% in identifying persons with mild NCD [67].

#### 4.1.5. Advantages

The main advantage of VR-based cognitive assessments is the potential for testing visuospatial reasoning in 3D, a key indicator of AD- and major-NCD-related cognitive decline that traditional tests fail to fully account for [68]. In addition to this, using computer-based testing allows for streamlined cloud-storage of data and greatly decreases variability in test administration; tests can even be administered by the patients themselves if they have the right equipment [67]. As the tests are easily administered, data from multiple trials can be gathered, improving the accuracy and sensitivity/specificity of classifiers [66,67]. This also allows for more complex queries to be readily addressed in a controlled environment that would not be possible or feasible to do in a real environment, and the parameters of that environment can be readily changed as needed [69]. Additionally, the “gamification” of the testing task can counter the anxiety of standard testing, changing the activity into a more relaxed, engaging experience [65].

Neuropsychological tests also have limited relation to an individual’s ability to successfully complete activities of daily living, which is important for ascertaining the capabilities of a person and whether they can live independently [70]. VR tools have been found to relate strongly to their real-world counterparts, particularly in navigation [57], and can be used to test certain facets such as episodic memory in a more realistic, utilitarian manner [59]. VR tests have superior ecological validity compared to their pen-and-paper counterparts [64,65]. Such tests of instrumental activities of daily living are invaluable in the diagnosis of pre-major NCD cases, and can help to predict the conversion of mild NCDs into AD [64]. VR also aids in the development of dual task tests that assess both one’s cognition and ability to maintain gait.

#### 4.1.6. Disadvantages

One disadvantage of VR-based diagnosis is that there is a relatively higher fixed cost than other tests for purchasing the required equipment. The practicality of VR-based tests and treatments improves, however, as the cost of VR headsets and equipment decreases with time [71]. VR-based tests also require more expertise to administer than current tests. In addition, one aspect of cognition in which VR may be limited is the evaluation of social cues. Traditional cognitive tests involve a conversation between an administrator and patient, an aspect that is lost in VR. Adaptations of current cognitive assessment tools such as the MMSE to VR may yield results that are incompatible with those of the original versions simply because of loss of direct verbal communication between administrator and patient. In addition, some users of VR systems have problems with motion sickness, which can hinder their performance and can make use of the VR system unpleasant [51]. Tests such as the “Simulator Sickness Questionnaire” can allow for the identification of those who are susceptible to experiencing discomfort from the VR environment in a way that may alter their results [72]; certain features of the VR experience, such as many graphical frames per second and limitation of sharp turns, can be tuned to mitigate discomfort [65]. General technophobia is also an issue that can impede the use of VR and other computerized tests [64,65]. Some of these issues in accessibility, such as motion sickness, appear to be alleviated in recent work, perhaps due to improvements in the technology [73]. As VR technology is made more advanced and becomes more commonplace, these disadvantages may be mitigated further.

### 4.2. Virtual-Reality-Based Therapy

#### 4.2.1. Background

There is currently no known pharmaceutical treatment or preventive intervention for major NCDs or mild NCDs. Certain behaviors and lifestyle choices can reduce a person’s risk for developing cognitive impairment, however, and there is evidence that AD is inversely correlated with mental stimulation and education [74]. Thus, as the search for an effective drug treatment continues, it may be worthwhile to invest in the development of mental exercises that may be able to slow the progression of cognitive decline. The use of computer environments to benefit cognitive capacity has been found to be useful, and has been employed in the context of mild NCDs. Drills for training certain aspects of cognition, specially developed videogames, dubbed “serious games”, and virtual reality scenarios fall under the umbrella of computerized cognitive training. Though not necessarily taking place in a VR environment, such programs indicate that gamified exercises have considerable utility. Cognitive benefits, alongside certain emotional and well-being improvements, have been observed following computerized cognitive training sessions [75,76,77]. Both specially designed and commercially available (e.g., Nintendo Wii) “exergames” have been found to improve balance in the older population [78,79]. Serious games such as “Kitchen and Cooking”, which involve users completing cognitively taxing tasks or emulating real-life situations that are important to daily living, are generally well received by the older population and have significant benefits to cognition and independence [80]. VR games are especially promising in the training of 3D navigation. In addition, a major risk factor for development of major NCDs is depression. One potential function of VR is the ability to immerse patients in alternate worlds with the goal of improving mood.

#### 4.2.2. Virtual Reality Exercise

Lee et al. designed a 12-week virtual reality exercise program for mild NCD patients. Patients were brought to three gaming sessions per week, for a total of 36 sessions. During this time, balance, depression, and quality of life were tracked in the patients, and patients demonstrated significant improvement across all three metrics. A control group enrolled in traditional therapy sessions saw no significant improvement [81]. Hwang et al. utilized a VR training program on 24 older persons with mild NCD, finding that balance and cognitive test scores were improved relative to controls, though details of the VR training program used are unclear [82]. Htut et al. found that a virtual reality video-game-based exercise, which is less beneficial than standard physical activity in assessments of physical ability but better than controls, can also simultaneously improve cognitive function [83]. Integration of virtual environments with existing exercise modalities, such as stationary cycling, has been found to be appropriate for younger and older individuals alike, while being rated as enjoyable by users, promoting long-term adherence to the exercise program [81,84,85].

#### 4.2.3. Virtual Cognitive Exercises

As VR tests are effective in evaluating the navigational abilities of a patient, they may prove useful for rehabilitating those abilities. Kober et al. applied a VR navigation simulation based on Graz, Austria. Participants did not have mild NCDs or Alzheimer’s disease, but rather exhibited brain damage with notable deficiencies in spatial orientation. Users’ general performance on spatial cognitive tests was found to improve following the VR training program [86]. Regarding cognitive disorders, Man et al. found that VR-based memory training exhibited greater improvements in memory compared to taking part in therapist-led memory training [87]. However, Fasilis et al. found little significant improvement in major NCD patients engaged in serious VR games, though in this study, the total time participants spent with the game was limited: The training regimen was 10 h over a 4–5 week period [88].

#### 4.2.4. Dual-Task Training

Dual-task training involves the performance of a cognitively demanding task while the subject is walking. Such tasks are useful for the testing of gait control, the loss of which is typical of mild NCDs, and especially in cases of Parkinson’s disease [89]. Liao et al. created a dual-task VR exercise program incorporating both physical and cognitive training, and found that the program improved executive function and gait performance comparable to a combined physical and cognitive training regimen, but the dual-task performance was significantly improved in the VR group relative to the non-VR group. The unique ability VR systems have in combining physical activity with an interactive environment appears to be a significant boon to addressing deficiencies in dual-task performance [90].

#### 4.2.5. Advantages

VR-based therapy allows patients to train cognitive function without supervision, and users can potentially train in their own homes provided they have the right equipment [91]. In addition, they give the potential for patients to practice daily tasks with minimal risk. Patient performance data on tasks can be easily stored in the cloud for analysis. Furthermore, users of some VR training programs describe them as “fun”, which can encourage adherence to a VR-based cognitive training program [82], and VR exercise programs are more well received than standard physical activity regimens [83]. As noted previously, virtual reality exercise allows for the easy integration of a cognitive component into the exercise [92], which can provide tandem physical and cognitive benefits that perhaps take advantage of the promotion of neuroplastic processes induced by exercise [92,93]. The feeling of immersion that VR brings with it also seems to be an important factor in improving spatial abilities. The effects of immersive (VR) and non-immersive versions of the game “Fruit Ninja” were studied on 33 persons that averaged 62 years old over a four-week period. The VR exergaming participants experienced improvements in inhibition and task switching that the non-VR group did not [94].

#### 4.2.6. Disadvantages

Again, there is a high fixed cost for initial purchase of equipment. Although VR would potentially enable patients to train their mental faculties independently of clinicians, in practice, perhaps the VR equipment would be too expensive for most people to invest in at home; thus, the responsibility of guiding the training still falls to the physician. In addition, perhaps VR is limited in its ability to train social faculties. Furthermore, as mentioned previously, some individuals experience motion sickness from virtual environments, and the discomfort may limit or prevent their use of therapeutic VR [51]; though the aforementioned studies note that participants largely tolerated and enjoyed VR systems, sample sizes were relatively small; larger studies with more rigorously explained inclusion and exclusion criteria are necessary for further analysis of the prevalence of tolerability issues. Furthermore, serious games in general are considered to be most adapted to those with more mild NCDs, rather than those with AD and major NCDs; those with major NCDs also require the assistance of a caregiver more so than those with mild NCDs [95].

### 4.3. Augmented and Mixed Reality

Augmented reality (AR) differs from virtual reality in that it involves the insertion of virtual elements into the real environment, rather than the creation of a fully virtual space. It typically involves an indirect display of the real environment, with virtual information and features overlaid onto this projection, thereby enhancing a user’s sense of reality. “True” AR is largely limited to displays of information and non-interactive elements, with the addition of interactive, completely virtual elements falling under the umbrella of mixed reality (MR). Mixed reality exists between VR and AR on the spectrum of reality, with virtual, interactable elements introduced into a display of the real world [96]. For both MR and AR applications, cameras and other tracking mechanisms are necessary for recognizing what is in the user’s environment and how the user is interacting with it; systems can be stationary or mobile, though the computing power of mobile systems can be limited. AR applications in daily life are somewhat limited by the “fashion” and social acceptability of these systems. The computing power necessary for complex interactions can necessitate a noticeably bulky system, and interactions through speech and gestures can be somewhat disruptive or awkward to perform in public [97]. The near ubiquity of mobile devices and the advent of cloud computing provide new opportunities for readily available AR, and mobile augmented reality (MAR) systems have been employed in a variety of applications, such as navigation, education, and entertainment [98].

AR systems have been shown to enhance learning, though studies showing this have largely involved relatively young students rather than older and/or cognitively impaired individuals [99,100]. Mobile augmented reality systems using one’s phone have also been proposed for the gamification of certain activities, such as maintaining hydration in the older population [101].

#### Augmented and Mixed Reality for AD and Major NCDs

There have been several AR applications developed with AD patients in mind. Rohrbach et Al. investigated an AR assistant for activities of daily living. The researchers utilized a Microsoft HoloLens and their application, TherapyLens, and tested AD patients’ abilities to make a cup of tea. The program featured holographic cues that indicated the steps required. Through voice commands, users could advance through the steps. The ultimate utility of the program was unclear, as those with the AR assistance did not have improved performance, and their time to complete for each task was actually increased. Researchers stated that some participants had difficulty remembering how to command the system, noting that the impaired ability to learn in patients of neurodegenerative disorders may make such AR systems difficult to use. Though participants expressed interest in the AR system, some noted that the system was uncomfortable, and others did not notice or remember the holograms [102]. Other systems, such as cARe, have been proposed, using AR projections to help instruct major NCD patients on activities of daily living. cARe notably includes the ability to “time out”, which again prompts the user and provides instruction for the current step, which can be helpful if the patient becomes distracted or forgets their task. Praise-based incentives were also included to motivate patients. Studies for cARe are still ongoing, though initial results suggest cARe is more helpful in a cooking scenario than written instructions. However, as with TherapyLens, problems arose from the voice communication, and the system did not account for patients altering the order of steps [103].

Aruanno et al. created a HoloLens-based MR cognitive trainer. Two prototypes of the trainer were tested; one was gesture controlled, while the second, which was modified based on feedback from the first prototype, was both voice and gesture controlled. Both featured various text-instructed activities geared toward short-term and spatial memory training, involving the location of certain objects in boxes projected throughout the user’s environment. Though controls were not included in the study, participants notably found the trainer engaging, and the second prototype was easy to use even for participants unfamiliar with technology. There was minimal discomfort reported; the HoloLens itself is designed to reduce the motion-sickness and fatigue that are typical of VR systems. Furthermore, by foregoing the full immersion of VR, MR allows the user to remain aware of the real environment. Feedback was positive overall, and the study illuminates some concerns that need to be addressed in the development of MR cognitive trainers, such as where and how to place textual instructions, what feedback (sound, visual, etc.) to provide the user, and interaction methods [104]. Park et al. observed the effects of an MR cognitive trainer using the Consortium to Establish a Registry for Alzheimer’s Disease, ultimately finding the program to be a significant improvement over conventional cognitive training in improving visuospatial working memory [105].

## 5. Conclusions

Although pharmaceutical treatment is not yet available for major NCDs, there is still utility in the early diagnosis of the condition for both assisting patients in managing it, as well as for aiding in further research that may produce an effective treatment. In addition, non-pharmacological therapy, such as serious games and VR-based cognitive training, can serve as an effective means of improving and preserving cognitive functions. Currently, early diagnosis can facilitate decision-making regarding a patient’s current and future assistive care options. Thus, there is a need for an effective cognitive test for major NCDs. Both the pragmatism and the scientific merit of a test must be taken into consideration. The most convenient cognitive test is perhaps the clock-drawing test, though this is lacking in its testing of language abilities. The test that currently has the greatest utility is perhaps the MMSE, which, due to its popularity, has become the standard for use in cognitive studies. However, due to its associated intellectual property issues, as well as its deficiencies in testing visuospatial capabilities, perhaps a more extensive assessment such as the ACE should be considered to maximize accuracy, sensitivity, and specificity. To resolve the intellectual property issues, perhaps a simple test with no associated patent, such as the GPCOG, would be of use. However, to date, no test in common use is able to assess visuospatial reasoning in 3D, a major deficit of amnestic mild NCDs and AD.

Thus, perhaps there is much potential for the incorporation of VR-based 3D environments into the diagnosis of major NCDs, as well as in the training of mental faculties and alleviation of depression. As the technology powering virtual, augmented, and mixed realities becomes more accessible, it becomes more feasible to employ the technologies in both clinical and at-home settings. We propose that VR spatial tests such as virtual mazes be developed and studied, as the importance of visuospatial reasoning ability for distinguishing one’s major NCD state is apparent. Such VR programs could serve both as tests and as cognitive trainers themselves. AR and MR approaches may be optimal for home use, as they allow the user to maintain environmental awareness, while still allowing for a 3D spatial component to games and activities. ADL tasks also present a test that is relatively easy to understand and carry out and that can help determine the level of one’s cognitive impairment. The increased engagement noted with AR and MR activities can incentivize mild and major NCD patients to perform tasks that are vital to preserving their mental faculties.

## Figures and Tables

**Table 1 jcm-09-03287-t001:** Summary of advantages and disadvantages of cognitive tests.

Test Name	Advantages	Disadvantages
Mini-Mental State Examination	Popular, so there is a large pool of available testing results. Simple and relatively short. High accuracy and specificity, with moderate sensitivity [22]. Assesses a large range of cognitive functions.	Intellectual property issues create financial barriers for clinicians. Limited testing of visuospatial reasoning. Heavily language-based, making results dependent on the subject’s overall language ability [27,28,29,30].
Clock-Drawing Test	Simple and easy to administer. Short time to administer. Assesses visuospatial reasoning	Open-ended, making scoring inconsistent. Can be inaccurate on its own. Does not test language. Results correlate with education level [35].
Addenbrooke’s Cognitive Examinations	Easy to administer. Tests visuospatial capability. More sensitive than MMSE, with lower impact of ceiling and floor effects [37]. Effective even when translated [39].	As it contains portions of the MMSE, there are issues related to intellectual property. Some questions assume patient knowledge of certain subjects.
General Practitioner Assessment of Cognition	Short, but tests many different areas of cognition [41]. Can be taken in written and online formats. Agnostic to cultural background and education level [42].	Requires input from an informant. Apparent correlation between scores and age.
Montreal Cognitive Assessment	Short, but tests many different areas of cognition. Includes multiple tests of visuospatial capacity. Freely available and easy to administer. Short time to administer	Sensory impairments can impact score [49]. Failure rates of questions vary considerably [50].

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
