# Peer review of "Current Cognition Tests, Potential Virtual Reality Applications, and Serious Games in Cognitive Assessment and Non-Pharmacological Therapy for Neurocognitive Disorders"

_jcm, 2020, doi:10.3390/jcm9103287_

Round 1

Reviewer 1 Report

Nice  review about the current situation but probably too long and not clearly separating the new perspectives and the current situation.

Author Response

“Nice  review about the current situation but probably too long and not clearly separating the new perspectives and the current situation.”

Response:  We have shortened the introduction considerably, and have made it more clear what the current state of VR specifically with regard to cognitive impairment.

Reviewer 2 Report

This review examines the potential applications for VR, AR and mixed reality technologies to assess and treat cognitive deficits associated with Alzheimer’s dementia and MCI.  The authors summarize the advantages and disadvantages of clinically used assessment tools, and proceed to discuss how VR cognitive assessment may address unaddressed needs in current assessment methods.   They then provide a brief review of VR therapy for addressing dementia related cognitive deficits.

The manuscript is well written, and in addressing an emerging technology that is increasingly used in psychiatry and neurology, is on a timely topic.  However, I have some suggestions and considerations for the manuscript:

  1. Section 2.1.2 Diagnostic Needs:  “the test should be agnostic to the administrator and rater” – the meaning of this is unclear.
  2. Section 2.2 Areas of Cognition: The authors do not list some important and commonly considered domains of cognition, such as processing speed, executive functioning - and within the domain of attention - selective and sustained attention.  Given the heterogeneity in the literature around defining cognitive domains, could the authors please specify why they chose to highlight the domains that they did (i.e. are these the ones that are considered particularly pertinent to dementia/MCI?). 
  3. Section 2.2 Areas of Cognition: The authors list “emotion – the ability to experience alterations in mood” as a cognitive area.  The experience of affect itself is not generally considered a cognitive domain, although certain aspects of social cognition may be considered to have an affective component (identifying facial emotional expressions in other, theory of mind).
  4. Section 3 – Common Cognitive Assessments: The authors do not discuss the MoCA, which at least in North America is very commonly used in clinical settings.
  5. Section 4.1.5 Advantages of VR, line 385: The authors very correctly discuss the limited correlation between traditional neuropsychological test performance and ADLs.  I would see this as a critical point that bears emphasizing, and one of the major arguments (in addition to its ability to assess visuospatial reasoning) for considering VR technologies.  I would suggest discussing the correlation (or lack thereof) between real world functioning and currently used cognitive assessments in more detail in Section 3, and perhaps also adding a section on currently used methods of assessing functional capacity/ADLs in clinical practice.   This would help the reader grasp this as a current unmet need, and strengthen the rationale for incorporating VR.
  6. Section 4.1.5 Disadvantages: The authors discuss some of the tolerability issues with VR, such as motion sickness.  It would be helpful if the authors could include some of the data regarding user engagement/tolerability from the studies that they included which used VR to assess cognition.
  7. Section 4.2.1 Background on Virtual Reality-based therapy: “The use of computer environments to benefit cognitive capacity have been found to be useful, particularly in instances of mild cognitive impairment.” Could the authors please provide a reference for this.
  8. Section 4.2 Virtual Reality-based Therapy: In this section, I would recommend focusing only on studies involving participants with MCI or dementia.  The differences in psychopathology between dementia and stroke/brain injury do not allow for generalization of results. 
  9. Section 5 Conclusion: The authors mention here, and in the abstract, the use of VR for “alleviation of depression”.  However, no data is presented in the manuscript suggesting that VR therapy is useful for depressive symptoms in MCI/depression.

Author Response

This review examines the potential applications for VR, AR and mixed reality technologies to assess and treat cognitive deficits associated with Alzheimer’s dementia and MCI.  The authors summarize the advantages and disadvantages of clinically used assessment tools, and proceed to discuss how VR cognitive assessment may address unaddressed needs in current assessment methods.   They then provide a brief review of VR therapy for addressing dementia related cognitive deficits.

The manuscript is well written, and in addressing an emerging technology that is increasingly used in psychiatry and neurology, is on a timely topic.  However, I have some suggestions and considerations for the manuscript: 

  1. Section 2.1.2 Diagnostic Needs:  “the test should be agnostic to the administrator and rater” – the meaning of this is unclear.

Response: We have reworded this to make it clearer.

  1. Section 2.2 Areas of Cognition: The authors do not list some important and commonly considered domains of cognition, such as processing speed, executive functioning - and within the domain of attention - selective and sustained attention.  Given the heterogeneity in the literature around defining cognitive domains, could the authors please specify why they chose to highlight the domains that they did (i.e. are these the ones that are considered particularly pertinent to dementia/MCI?). 

Response: We have opted to remove section 2.2.

  1. Section 2.2 Areas of Cognition: The authors list “emotion – the ability to experience alterations in mood” as a cognitive area.  The experience of affect itself is not generally considered a cognitive domain, although certain aspects of social cognition may be considered to have an affective component (identifying facial emotional expressions in other, theory of mind).

Response: We have opted to remove section 2.2

  1. Section 3 – Common Cognitive Assessments: The authors do not discuss the MoCA, which at least in North America is very commonly used in clinical settings.

Response: We have added a section on the MoCA

  1. Section 4.1.5 Advantages of VR, line 385: The authors very correctly discuss the limited correlation between traditional neuropsychological test performance and ADLs.  I would see this as a critical point that bears emphasizing, and one of the major arguments (in addition to its ability to assess visuospatial reasoning) for considering VR technologies.  I would suggest discussing the correlation (or lack thereof) between real world functioning and currently used cognitive assessments in more detail in Section 3, and perhaps also adding a section on currently used methods of assessing functional capacity/ADLs in clinical practice.   This would help the reader grasp this as a current unmet need, and strengthen the rationale for incorporating VR.

Response: We have added information on current assessments of ADL (which consists primarily of patient/caretaker questionnaires) in section 4, in order to keep section 3 focused on the more standardized cognitive tests.

  1. Section 4.1.5 Disadvantages: The authors discuss some of the tolerability issues with VR, such as motion sickness.  It would be helpful if the authors could include some of the data regarding user engagement/tolerability from the studies that they included which used VR to assess cognition.

Response: The reported studies reported the systems were tolerated and enjoyed by users, but due to small sample sizes and lack of clarity on some inclusion/exclusion criteria, it is difficult to judge the prevalence of the issues. We have made a note of this in the text.

  1. Section 4.2.1 Background on Virtual Reality-based therapy: “The use of computer environments to benefit cognitive capacity have been found to be useful, particularly in instances of mild cognitive impairment.” Could the authors please provide a reference for this.

Response: This statement refers broadly to the results we present in the section as a whole. However, we see that the wording could be misinterpreted as the benefits are exclusive to MCI; we have rephrased this statement to clarify this.

  1. Section 4.2 Virtual Reality-based Therapy: In this section, I would recommend focusing only on studies involving participants with MCI or dementia.  The differences in psychopathology between dementia and stroke/brain injury do not allow for generalization of results. 

Response: We included VR programs outside the scope of dementia/MCI as demonstrations of the concept of VR assessment that accounts for multiple areas of cognition. We have made this clearer.

  1. Section 5 Conclusion: The authors mention here, and in the abstract, the use of VR for “alleviation of depression”.  However, no data is presented in the manuscript suggesting that VR therapy is useful for depressive symptoms in MCI/depression.

Response: Though alleviation of depression was not a focus of our work, a reference we cited, entitled “Virtual Reality Exercise Program on Balance, Emotion and Quality of Life in Patients with Cognitive Decline” observed the effects of a VR exercise program, which we noted in the manuscript in section 4.2.2.

Reviewer 3 Report

This article aims to review the current cognition tests and the potential applications of virtual reality for the assessment and non-pharmacological therapy for patients with neurocognitive disorder. Because of concerned related (partly) below,

General comments:

Please prefer using "major neurocognitive disorder" instead of "dementia" and "mild neurocognitive disorder" instead of "mild cognitive impairment", that would be closer from current clinical semantics.

Because terms like seniors, elderly, the aged, aging dependents, old-old, young-old, and similar “other-ing” terms connote a stereotype, avoid using them. Terms such as older persons, older people, older adults, older patients, older individuals, persons 65 years and older, or the older population should be preferred.

Specific comments:

The introduction should be drastically shortened. Indeed, the aims of this review is not to describe what is a major or minor neurocognitive disorder and what is the pathological process for Alzheimer disease. One short paragraph summarizing these aspects and referring to another recent review (less than five years) focuses on the topic would be more adapted, and would give an introduction more focused on the topic (cognitive assessment and VR). Moreover, even the review is not systematic, a short paragraph explaining the method used to found (database and main keywords), and to select (date?, age?, exclusion for some topics?) articles, as the informative number of article found would help readers to understand your choices, especially for the last part concerning the VR (which is the core part of the review).

The first paragraph of the part 2 "Motivation for Cognitive Assessment" should adopt a more clinical point of view. Indeed, the first need for cognitive assessment is the clinical need to understand if there really is a neurocognitive disorder (or if the diagnosis is related to depression, anxiety etc...), and which cognitive functions are affected. Regarding the DSM V diagnostic criteria or the IWG-2 research criteria proposed by Dubois et al, the cognitive assessment is a core clinical criterion. This is the most important "Motivation for Cognitive Assessment".

Part 2 is particularly important and accurate in this review. Nevertheless, no reference is provided. References must be provided in this important theoretical part of the review. Moreover, part 2.2 was probably included because of the descriptions of the cognitive functions provided by the tests described in part 3, but it is not necessary in my opinion. Nevertheless, if the authors want to keep this part 2.2, it should be reworked because of some approximations probably due to the simplification proposed by the authors. Line 154, point 2 is not necessary and should be added before point 3 line 155, 4 line 157 and 5 line 159. As described line 163, calculation is not a basic cognitive function but is based on other cognitive functions from different cognitive process that are basic. the aim of a cognitive assessment is to test those basic functions (by using calculation for example). Thus, in my opinion, only the basic function should appear in this section 2.2 e.g visual and auditive input, central administrator, working memory, executive functions (attention, concentration, switch, reasoning), semantic memory, episodic memory etc...

Part 3: I think the Montreal Cognitive Assessment (MoCA) should be added as it is a widely used test.

Table 1: a fourth column should be added between the first and the second column to isolate and to detail the cognitive properties of the tests.

line 178-179: praxis and visuo-spatial abilities are also assessed in the MMSE ("Take a paper in your right hand, fold it in half, and put it on the floor” assess not only language but also praxis, and the copy design also assess praxis and visuo-spatial abilities).

line 183: please explain the abbreviation 3MS

line 202: IP should not be abbreviated as it appears only twice in the text and two times in the table one.

line 206-206: the intellectual property is exact, but in practice, this test is so universally known that for regular users, no support is needed, and not patent is paid. Thus, this argument should be minimized in the text.

line 206-209: the statement is not exact as the last question of the MMSE explore (indeed very briefly) visuospatial abilities. Please, make the text more open on this point.

line 213: please explain the abbreviation COMPASS (COmputational Model to Predict the development of Alzheimer’s diSease Spectrum).

line 248: please add (ACE) just after "Addenbrooke's Cognitive Examination"

line 312: could you please add some other cognitive abilities, using bracket for example.

part 4: there is a misleading between serious (exer)game in virtual environment and virtual reality. Most of the article cited are describing serious (exer)gaming in a virtual environment and not VR (immersive experience using VR headset or VR room with large immersive screens), but authors do not differentiate the two technologies, which can lead to misevaluate the current place of VR in the assessment and non-pharmacological treatment for patients with neurocognitive disorder. Moreover, you cited at least one retracted article: doi:10.1097/MD.0000000000014752.

line 413: the authors must remove "prevention" in this sentence. Indeed, there are numerous consistent data that prove the positive impact of regular physical activity, low alcohol consumption and "Mediterranean" diet to decrease the incidence of neurocognitive disorders.

line 415: please use a more consensual word instead of "miracle drug".

line 425: you could also cite the serious exergame X-torp (https://www.youtube.com/watch?v=owI_IBvPMz0&t=50s); Ben-Sadoun G, Sacco G, Manera V, et al. Physical and Cognitive Stimulation Using an Exergame in Subjects with Normal Aging, Mild and Moderate Cognitive Impairment. J Alzheimers Dis. 2016;53(4):1299-1314. doi:10.3233/JAD-160268

Author Response

This article aims to review the current cognition tests and the potential applications of virtual reality for the assessment and non-pharmacological therapy for patients with neurocognitive disorder. Because of concerned related (partly) below,

General comments:

Please prefer using "major neurocognitive disorder" instead of "dementia" and "mild neurocognitive disorder" instead of "mild cognitive impairment", that would be closer from current clinical semantics.

Because terms like seniors, elderly, the aged, aging dependents, old-old, young-old, and similar “other-ing” terms connote a stereotype, avoid using them. Terms such as older persons, older people, older adults, older patients, older individuals, persons 65 years and older, or the older population should be preferred.

Response: We have updated the terminology used in the work.

Specific comments:

The introduction should be drastically shortened. Indeed, the aims of this review is not to describe what is a major or minor neurocognitive disorder and what is the pathological process for Alzheimer disease. One short paragraph summarizing these aspects and referring to another recent review (less than five years) focuses on the topic would be more adapted, and would give an introduction more focused on the topic (cognitive assessment and VR). Moreover, even the review is not systematic, a short paragraph explaining the method used to found (database and main keywords), and to select (date?, age?, exclusion for some topics?) articles, as the informative number of article found would help readers to understand your choices, especially for the last part concerning the VR (which is the core part of the review).

Response: We have summarized the sections on MCI and AD and shortened the introduction considerably.

The first paragraph of the part 2 "Motivation for Cognitive Assessment" should adopt a more clinical point of view. Indeed, the first need for cognitive assessment is the clinical need to understand if there really is a neurocognitive disorder (or if the diagnosis is related to depression, anxiety etc...), and which cognitive functions are affected. Regarding the DSM V diagnostic criteria or the IWG-2 research criteria proposed by Dubois et al, the cognitive assessment is a core clinical criterion. This is the most important "Motivation for Cognitive Assessment".

Response: We have included more information on the diagnostic standards.

Part 2 is particularly important and accurate in this review. Nevertheless, no reference is provided. References must be provided in this important theoretical part of the review. Moreover, part 2.2 was probably included because of the descriptions of the cognitive functions provided by the tests described in part 3, but it is not necessary in my opinion. Nevertheless, if the authors want to keep this part 2.2, it should be reworked because of some approximations probably due to the simplification proposed by the authors. Line 154, point 2 is not necessary and should be added before point 3 line 155, 4 line 157 and 5 line 159. As described line 163, calculation is not a basic cognitive function but is based on other cognitive functions from different cognitive process that are basic. the aim of a cognitive assessment is to test those basic functions (by using calculation for example). Thus, in my opinion, only the basic function should appear in this section 2.2 e.g visual and auditive input, central administrator, working memory, executive functions (attention, concentration, switch, reasoning), semantic memory, episodic memory etc...

Response: We have opted to remove section 2.2.

Part 3: I think the Montreal Cognitive Assessment (MoCA) should be added as it is a widely used test.

Response: We have added a section on the MoCA.

Table 1: a fourth column should be added between the first and the second column to isolate and to detail the cognitive properties of the tests.

Response: We have made it more clear what aspects are assessed by each text in the text, but adding another column to the table would make it considerably more difficult to read.

line 178-179: praxis and visuo-spatial abilities are also assessed in the MMSE ("Take a paper in your right hand, fold it in half, and put it on the floor” assess not only language but also praxis, and the copy design also assess praxis and visuo-spatial abilities).

Response: We have changed the wording to make it more clear that assessment of visuospatial ability is limited rather than nonexistent.

line 183: please explain the abbreviation 3MS

Response: We have added the full name (modified mini-mental state exam).

line 202: IP should not be abbreviated as it appears only twice in the text and two times in the table one.

Response: We have expanded all instances of “IP” to “intellectual property.”

line 206-206: the intellectual property is exact, but in practice, this test is so universally known that for regular users, no support is needed, and not patent is paid. Thus, this argument should be minimized in the text.

Response: We have changed the wording to lessen its importance, but have still included it as obtaining the official, up-to-date version must be paid for.

line 206-209: the statement is not exact as the last question of the MMSE explore (indeed very briefly) visuospatial abilities. Please, make the text more open on this point.

Response: We have changed the wording to make it more clear that assessment of visuospatial ability is limited rather than nonexistent.

line 213: please explain the abbreviation COMPASS (COmputational Model to Predict the development of Alzheimer’s diSease Spectrum).

Response: We have elaborated on the abbreviation in the text.

line 248: please add (ACE) just after "Addenbrooke's Cognitive Examination"

Response: We have added the abbreviation where requested.

line 312: could you please add some other cognitive abilities, using bracket for example.

Response: This is deliberately open-ended, as the different programs could be designed to focus on certain cognitive aspects. We have reworded this to make it more clear.

part 4: there is a misleading between serious (exer)game in virtual environment and virtual reality. Most of the article cited are describing serious (exer)gaming in a virtual environment and not VR (immersive experience using VR headset or VR room with large immersive screens), but authors do not differentiate the two technologies, which can lead to misevaluate the current place of VR in the assessment and non-pharmacological treatment for patients with neurocognitive disorder. Moreover, you cited at least one retracted article: doi:10.1097/MD.0000000000014752.

Response: Thank you for pointing out the article retraction, we have removed it and its associated text. We have endeavored to restrict the non-VR serious games to the background in order to establish the (considerably larger) body of work that assess the value of gamified tasks in a virtual environment.

line 413: the authors must remove "prevention" in this sentence. Indeed, there are numerous consistent data that prove the positive impact of regular physical activity, low alcohol consumption and "Mediterranean" diet to decrease the incidence of neurocognitive disorders.

Response: We have changed “prevention” to “preventive intervention” and have noted that certain behaviors can reduce risk.

line 415: please use a more consensual word instead of "miracle drug".

Response: We have changed this to “effective drug treatment” and have also addressed other instances of overly emotive language.

line 425: you could also cite the serious exergame X-torp (https://www.youtube.com/watch?v=owI_IBvPMz0&t=50s); Ben-Sadoun G, Sacco G, Manera V, et al. Physical and Cognitive Stimulation Using an Exergame in Subjects with Normal Aging, Mild and Moderate Cognitive Impairment. J Alzheimers Dis. 2016;53(4):1299-1314. doi:10.3233/JAD-160268

Response: We have included the suggested reference, thank you for providing it.

Round 2

Reviewer 1 Report

well done, the text is much better to read and understand !

Author Response

Reviewer 1’s comments and authors’ responses

“well done, the text is much better to read and understand !”

Response: Thanks for your encouragement

Reviewer 3 Report

The revisions provided by the reviewer significantly improved the review, nevertheless, some minor revision should be made as suggested:

General comment: I am sorry to insist on this semantic point, but the correct word is neurocognitive disorder (NCD) end not cognitive disorder (CD). Please use the actual standard of the DSM 5.

Title: As the review includes not only articles concerning virtual reality but also virtual environment and serious games, the title should be modified. It could be something like “Current cognition tests, virtual reality and serious games: potential applications in older adults with neurocognitive disorder.” Independent of the title chooses, senile dementia should be replaced by older adult with neurocognitive disorder, or something equivalent.

Abstract:

Line 11: delate the word senile

Line 12-13: delate “, and AD-related major CD” as it is included in major neurocognitive disorder

Line 14: replace AD/ADRD by NCD

Main text

Line 63: replace MILD by mild

Line 317: replace MCI by mild NCD

Line 388: in this article, it is never clearly state that patients are in a VR environment and the pictures in the article are looking like virtual computerized environment but not really VR.

Line 395-401: same comment as line 388

Line 406: prefer using caregiver instead of caretaker

Line 501: prefer using virtual environment as the technology used in ref 84-86 are not really VR

Line 597: MR instead of VR

Line 607-8: you should include “serious games” in your sentence: “in addition, non-pharmacological therapy such as serious games and VR-based cognitive training…”

Line 630: please replace “mild CD, AD and ADRD patients” by “mild and major NCD patients”

Author Response

Reviewer 3’s comments and the authors’ responses

The revisions provided by the reviewer significantly improved the review, nevertheless, some minor revision should be made as suggested:

General comment: I am sorry to insist on this semantic point, but the correct word is neurocognitive disorder (NCD) end not cognitive disorder (CD). Please use the actual standard of the DSM 5.

Response: We have changed cognitive disorder (CD) to neurocognitive disorder (NCD)

Title: As the review includes not only articles concerning virtual reality but also virtual environment and serious games, the title should be modified. It could be something like “Current cognition tests, virtual reality and serious games: potential applications in older adults with neurocognitive disorder.” Independent of the title chooses, senile dementia should be replaced by older adult with neurocognitive disorder, or something equivalent.

Response: We have changed the title.

Abstract:

Line 11: delate the word senile

Line 12-13: delate “, and AD-related major CD” as it is included in major neurocognitive disorder

Line 14: replace AD/ADRD by NCD

Response: We have made the suggested alterations.

Main text

Line 63: replace MILD by mild

Line 317: replace MCI by mild NCD

Line 388: in this article, it is never clearly state that patients are in a VR environment and the pictures in the article are looking like virtual computerized environment but not really VR.

Line 395-401: same comment as line 388

Line 406: prefer using caregiver instead of caretaker

Line 501: prefer using virtual environment as the technology used in ref 84-86 are not really VR

Line 597: MR instead of VR

Line 607-8: you should include “serious games” in your sentence: “in addition, non-pharmacological therapy such as serious games and VR-based cognitive training…”

Line 630: please replace “mild CD, AD and ADRD patients” by “mild and major NCD patients”

Response: We have made the suggested changes and clarified where appropriate.